

# GANplifying event samples

Anja Butter[1], Sascha Diefenbacher[2*], Gregor Kasieczka[2],
Benjamin Nachman[3] and Tilman Plehn[1]

**1** Institut für Theoretische Physik, Universität Heidelberg, Germany
**2** Institut für Experimentalphysik, Universität Hamburg, Germany
**3** Physics Division, Lawrence Berkeley National Laboratory, Berkeley, CA, USA

⋆ sascha.daniel.diefenbacher@uni-hamburg.de

## Abstract

A critical question concerning generative networks applied to event generation in particle physics is if the generated events add statistical precision beyond the training sample. We show for a simple example with increasing dimensionality how generative networks indeed amplify the training statistics. We quantify their impact through an amplification factor or equivalent numbers of sampled events.

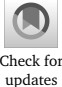
---

## Contents

---

## 1 Introduction

A defining feature of physics at the LHC is that we can compare a vast data set in all its details with first-principles predictions. The corresponding simulations start with a Lagrangian, compute the hard scattering in perturbative QCD, the parton shower in resummed QCD, model hadronization based on dedicated precision measurements, and finally add a full detector

simulation. Technically, all of this relies on Monte Carlo techniques. The upcoming high-luminosity run of the LHC will produce a data set more than 25 times the current Run 2 data set, with precision requirements seriously challenging the current simulation tools [1,2]. One way to speed up the simulations and, in turn, improve their precision is to employ modern machine learning.

A variety of generative machine learning models have been proposed, including well-studied methods such as generative adversarial networks (GAN) [3,4], variational autoencoders [5,6], and variations of normalizing flows [7,8]. This work will focus on GANs, the most widely studied approach in high energy physics so far. Fast precision simulation in particle physics starts with phase space integration [9,10], phase space sampling [11–13], and amplitude networks [14,15]. Especially interesting are NN-based event generation [16–20], event subtraction [21], detector simulations [22–30], or fast parton showers [31–34]. Deep generative models can also improve searches for physics beyond the Standard Model [35] or anomaly detection [36,37]. Finally, GANs allow us to unfold detector effects [38,39], surpassed only by the consistent statistical treatment of conditional invertible networks [40]. Roughly, these applications fall into two categories, (i) generative networks accelerating or augmenting Monte Carlo simulations or (ii) generative networks offering entirely new analysis opportunities. For the first kind of application the known big question is *how many more events can we sensibly GAN before we are limited by the statistics of the training sample*?

A seemingly straightforward and intuitive answer to this question is *as many examples as were used for training, because the network does not add any physics knowledge* [41]. However, there are reasons to think that a generative model actually contains more statistical power than the original data set. The key property of neural networks in particle physics is their powerful interpolation in sparse and high-dimensional spaces. It is also behind the success of the NNPDF parton densities [42] as the first mainstream application of machine learning to particle theory. This advanced interpolation should provide GANs with additional statistical power. We even see promising hints for network extrapolation in generating jet kinematics [18].

Indeed, neural networks go beyond a naive interpolation in that their architectures define basic properties of the functions it parameterizes. They can be understood in analogy to a fit to a pre-defined function, where the class of fit functions is defined by the network architecture. For example, some kind of smoothness criterion combined with a typical resolution obviously adds information to a discrete training data. If the network learns an underlying density from a data set with limited statistics, it can generate an improved data set up to the point where the network starts learning the statistical fluctuation and introduces a systematic uncertainty in its prediction. Ideally, this level of training defines a sweet spot for a given network, which we will describe in terms of an equivalent training-sample size. An extreme baseline which we will use in this paper is indeed the case where the true density distribution is known in terms of a few unknown parameters. With this information it is always better to fit those parameters and compute statistics with the functional form than to estimate the statistics directly from the data.

In the machine learning literature this kind of question is known for example as data amplification [43], but not extensively discussed. An interesting application is computer games, where the network traffic limits the available information and a powerful computer still generates realistic images or videos. This practical question leads to more formal question of sampling correctors [44]. Alternatively, it can be linked to a classification problem to derive scaling laws for networks fooling a given hypothesis test [43,45]. Unfortunately, we are not aware of any quantitaive analysis describing the gain in statistical power from a generative network. To fill this gap, we will use a much simpler approach, close to typical particle theory applications. If we know the smooth truth distribution, we can bin our space to define

quantiles — intervals containing equal probability mass — and compute the $\chi^2$-values for sampled and GANned approximations. Our toy example will be a camel back function or a shell of a multi-dimensional hypersphere, because it defines a typical resolution as we know it for example from the Breit-Wigner propagators of intermediate particles [19].

This paper is organized as follows: We introduce the network architecture as well as the framework and study a one-dimensional example in Sec. 2. In Sec. 3 we extend these results a two-dimensional ring, and in Sec. 4 to the shell of a 5-dimensional hypersphere. We conclude in Sec. 5.

## 2 One-dimensional camel back

The first function we study is a one-dimensional camel back, made out of two normalized Gaussians $N_{\mu,\sigma}(x)$ with mean $\mu$ and width $\sigma$,

$$P(x) = \frac{N_{-4,1}(x) + N_{4,1}(x)}{2} \ . \tag{1}$$

We show this function in Fig. 1, together with a histogrammed data set of 100 points. We choose this small training sample to illustrate the potentially dramatic improvement from a generative network especially for an increasing number of dimensions. As a benchmark we define a 5-parameter maximum-likelihood fit, where we assume that we know the functional form and determine the two means, the two widths and the relative height of the Gaussians in Eq. (1). We perform this fit using the IMINUIT [46] and PROBFIT [47] PYTHON packages. The correctly assumed functional form is much more than we can encode in a generative network architecture, so the network will not outperform the precision of this fit benchmark. On the other hand, the fit illustrates an optimal case, where in practice we usually do not know the true functional form.

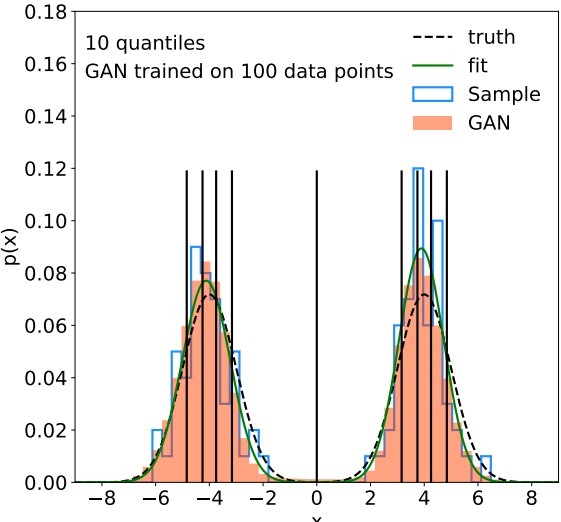

Figure 1: Camel back function as a 1D test case. We show the true distribution (black), a histogram with 100 sample points (blue), a fit to the samples data (green), and a high-statistics GAN sample (orange). Ten quantiles include 10% of the truth integral each.

To quantify the agreement for instance between the data sample or the fit on the one hand and the exact form on the other, we introduce 10, 20, or 50 quantiles. We illustrate the case of 10 quantiles also in Fig.1. We can evaluate the quality of an approximation to the true curve by computing the average quantile error

$$\text{MSE} = \frac{1}{N_{\text{quant}}} \sum_{j=1}^{N_{\text{quant}}} \left( x_j - \frac{1}{N_{\text{quant}}} \right)^2 \,, \tag{2}$$

where $x_j$ is the estimated probability in each of the $N_{\text{quant}}$ quantiles, which are defined with known boundaries. In a first step, we use this MSE to compare

1. low-statistics training sample vs true distribution;

2. fit result vs true distribution.

In Fig. 2 the horizontal lines show this measure for histograms with 100 to 1000 sampled points and for the fit to 100 points. For the 100-point sample we construct an error band by evaluating 100 statistically independent samples and computing its standard deviation. For the fit we do the same, *i.e.* fit the same 100 independent samples and compute the standard deviation for the fit output. This should be equivalent to the one-sigma range of the five fitted parameters folded with the per-sample statistics, if we take into account all correlations. However, we use the same procedure to evaluate the uncertainty on the fit, as is used for the other methods.

The first observation in Fig. 2 is that the agreement between the sample or the fit and the truth generally improves with more quantiles, indicated by decreasing values of the quantile MSE on the $y$-axis. which is simply a property of the definition of our quantile MSE error above. Second, the precision of the fit corresponds to roughly 300 hypothetical data points for 10 quantiles, 500 hypothetical data points for 20 quantiles, and close to 1000 hypothetical data points for 50 quantiles. This means that for high resolution and an extremely sparsely populated 1D-phase space, the assumed functional value for the fit allows the data to have the same statistical power as a dataset with no knowledge of the functional form that is 10 times bigger. If we define the *amplification factor* as the ratio between asymptotic performance to training events, the factor when using the fit information would be about 10. The question is,

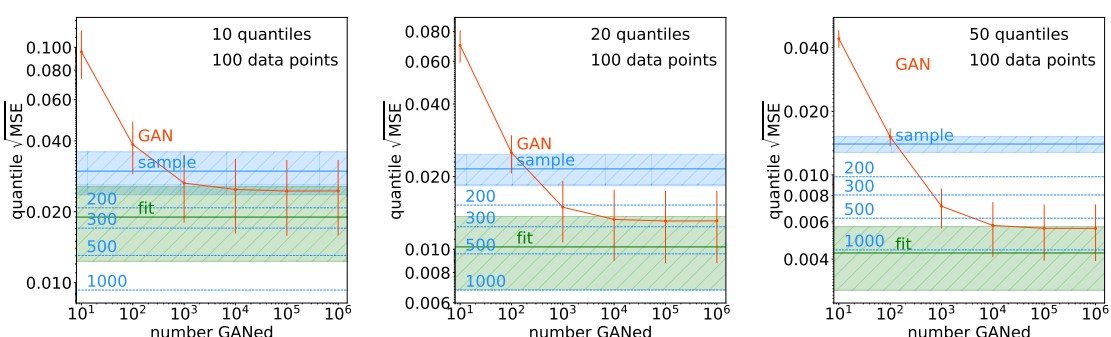

Figure 2: Quantile error for the 1D camel back function for sampling (blue), fit (green), and GAN (orange). We fit to and train on 100 data points, but also show (hypothetical) results for larger data sets with 200, 300, 500 and 1000 data points (dotted blue). These results were obtained using the same procedure as for the sample, but they have no influence on the GAN or fit. Left to right we show results for 10, 20, and 50 quantiles.

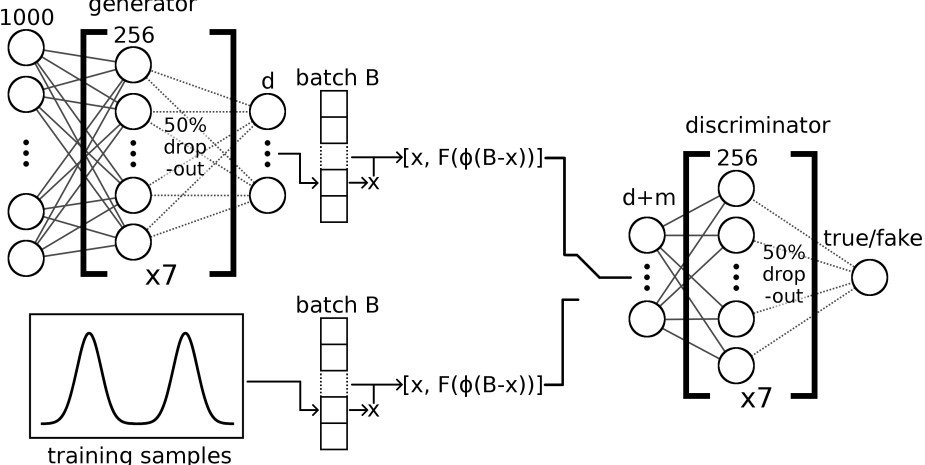

Figure 3: Diagram of our neural network architecture. The structure and usage of the embedding function $\Phi$, the aggregation function F, and other hyperparameters are described in the main text.

how much is a GAN with its very basic assumptions worth, for instance in comparison to this fit?

We introduce a simple generative model using the generator-discriminator structure of a standard GAN. This architecture remains generic in the sense that we do not use specific knowledge about the data structure or its symmetries in the network construction. Our setup is illustrated in Fig. 3. All neural networks are implemented using PYTORCH [48]. The generator is a fully connected network (FCN). Its input consists of 1000 random numbers, uniformly sampled from $[-1, 1]$. It is passed to seven layers with 256 nodes each, followed by a final output layer with $d$ nodes, where $d$ is the number of phase space dimensions. To each fully-connected layer we add a 50% dropout layer [49] to reduce over-fitting which is kept active during generation. The generator uses the ELU activation function [50].

The discriminator is also a FCN. In a naive setup, our bi-modal density makes us especially vulnerable to mode collapse, where the network simply ignores one of the two Gaussians. To avoid it, we give it access to per-batch statistics in addition to individual examples using an architecture inspired by DeepSets [51, 52]. This way its input consists of two objects, a data point $x \in \mathbb{R}^d$ and the full batch $B \in \mathbb{R}^{d,n}$, where $n$ is the batch size and $x$ corresponds to one column in $B$. First, we calculate the difference vector between $x$ and every point in $B$, $B - x$ with appropriate broadcasting, so that $B - x \in \mathbb{R}^{d,n}$ as well. This gives the discriminator a handle on the distance of generated points. This distance is passed to an embedding function $\Phi : \mathbb{R}^{d,n} \to \mathbb{R}^{m,n}$, where $m$ the size of the embedding. The embedding $\Phi$ is implemented as three 1D-convolutions (256 filters, 256 filters, $m$ filters) with kernel size 1, stride 1 and no padding. Each of the convolutions uses a LEAKYRELU [53] activation function with a slope of 0.01. For the embedding size we choose $m = 32$.

We then use an aggregation function $F : \mathbb{R}^{m,n} \to \mathbb{R}^m$ along the batch-size direction. The network performance is largely independent of the choice of aggregation function. Still we find that our choice of standard deviation slightly outperforms other options. We then concatenate $x$ and $F(\Phi(B-x))$ to a vector with length $d+m$. It is passed to the main discriminator network, an FCN consisting of a $d + m$ node input layer followed by seven hidden layers with 256 nodes each and a 1 node output layer. Mirroring the setup of the generator we once again intersperse each hidden layer with a 50% dropout layer. The discriminator uses 0.01 slope LEAKYRELU activation functions for the hidden layers and a SIGMOID function for the output.

During training we strictly alternate between discriminator and generator. Both networks

use the ADAM optimizer [54] with $\beta_1 = 0.5$, $\beta_2 = 0.9$ and a learning rate of $5 \times 10^{-5}$. This learning rate gets multiplied by 0.9 every 1000 epochs. The GAN is trained for a total of 10.000 epochs. To regularize the discriminator training we use gradient penalty [55] with $\gamma = 0.01$. Additionally, we apply a uniform noise ranging from $[-0.1, 0.1]$ to the training data in every discriminator evaluation, once again to reduce over-fitting and mode-collapse. We chose a batch size of $n = 10$ and the whole data set is shuffled each epoch, randomizing the makeup of these batches. One training with 100 data points takes approximately one hour on a NVIDIA TESLA P100 GPU. The architectures used for the different dimensionalities are identical except for changes to the generator output and discriminator input sizes.

Returning to the MSE-based performance comparison in Fig. 2, we now compare the three cases

1. low-statistics training sample vs true distribution;

2. fit result vs true distribution;

3. samples of GANned events vs true distribution.

The orange curve shows the performance of this GAN setup compared to the sample and to the 5-parameter fit. The GAN uncertainty is again derived by 100 independent trainings. We then compute the quantile error as a function of the number of GANned events and see how it saturates. This saturation is where GANning more events would not add more information to a given sample. Depending on the number of quantiles or the resolution this happens between 1000 and 10.000 GANned events, to be compared with 100 training events. This shows that it does make sense to generate more events than the training sample size.

On the other hand, training a GAN encodes statistical uncertainties in the training sample into systematic uncertainties in the network. This means that a GANned event does not carry the same amount of information as a sampled event. The asymptotic value of the GAN quantile error should be compared to the expected quantile error for an increased sample, and we find that the 10,000 GANned events are worth between 150 and 500 sampled events, depending on the applied resolution. Thus, our simple GAN produces an amplification factor above five for a resolution corresponding to 50 quantiles. An interesting feature of the GAN is that it follows the fit result with a slight degradation, roughly corresponding to the quoted fit uncertainty. In that sense the analogy between a fit and a flexible NN-representation makes sense, and the GAN comes shockingly close to the highly constrained fit.

## 3 Two-dimensional ring

In two dimensions we replace the camel back function by a Gaussian ring, namely

$$
\begin{aligned}
P(r) &= N_{4,1}(r) + N_{-4,1}(r), \\
P(\varphi) &= \text{const},
\end{aligned}
\tag{3}
$$

in polar coordinates with radius $r \geq 0$ and angle $\varphi$. The GAN architecture is the same as in the 1D case, with the input vector given in Cartesian coordinates $(x, y)$. This way the network has to learn the azimuthal symmetry in addition to the camel back shape when we slice through the ring and the origin. Technically, this generalization of the camel back function should be easier to learn for the GAN than the 1D case, because the region of high density is connected.

In the left and center panels of Fig. 4 we first show quantiles in the two polar coordinates separately, remembering that the network is trained on Cartesian coordinates. In our setup the GAN learns the peaked structure of the radius, with an amplification factor around four, much better than the flat distribution in the angle, with an amplification factor below two. Both of

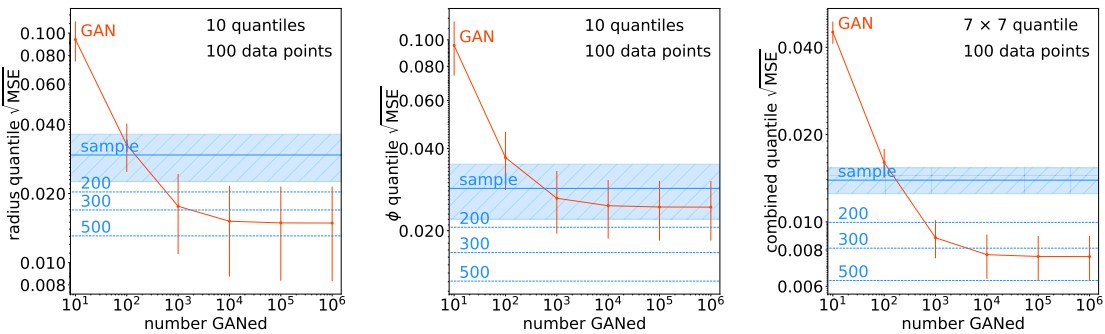

Figure 4: Quantile error for the 2D Gaussian ring for sampling (blue) and GAN (orange). Left to right we show 10 quantiles projected onto $r$ and $\varphi$, and $7 \times 7$ 2D-quantiles in polar coordinates.

these amplification factors are computed for ten quantiles, to be compared with the 1D-result in Fig. 2. We can combine the two dimensions and define $7 \times 7$ quantiles, to ensure that the expected number of points per quantile remains above one. The 2D amplification factor then comes out slightly above three, marginally worse than the 50 1D-quantiles shown in Fig. 2. One could speculate that for our simple GAN the amplification factor is fairly independent of the dimensionality of the phase space.

We illustrate the 49 2D-quantiles in Fig. 5, where the color code indicates the relative deviation from the expected, homogeneous number of 100/49 events per quantile. We see the effect of the GAN improvement with more subtle colors in the right panel. While it is hard to see the quality of the GAN in radial direction, we observe a shortcoming in the azimuthal angle direction, as expected from Fig. 4. We also observe the largest improvement from the GAN in the densely populated regions (as opposed to the outside) which agrees with the network learning to interpolate.

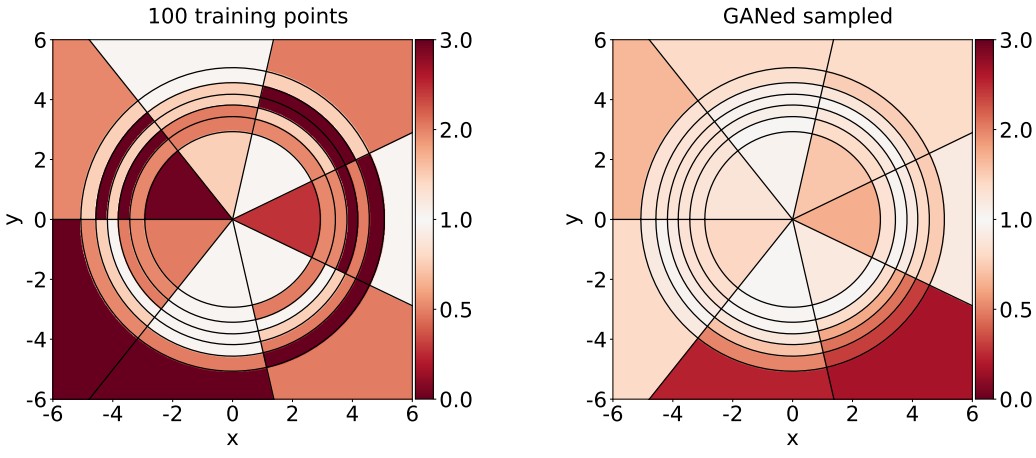

Figure 5: Relative deviation of the training sample (left) and the GANned events (right) for the 2D Gaussian ring. We show the same $7 \times 7$ 2D-quantiles as in Fig. 4,

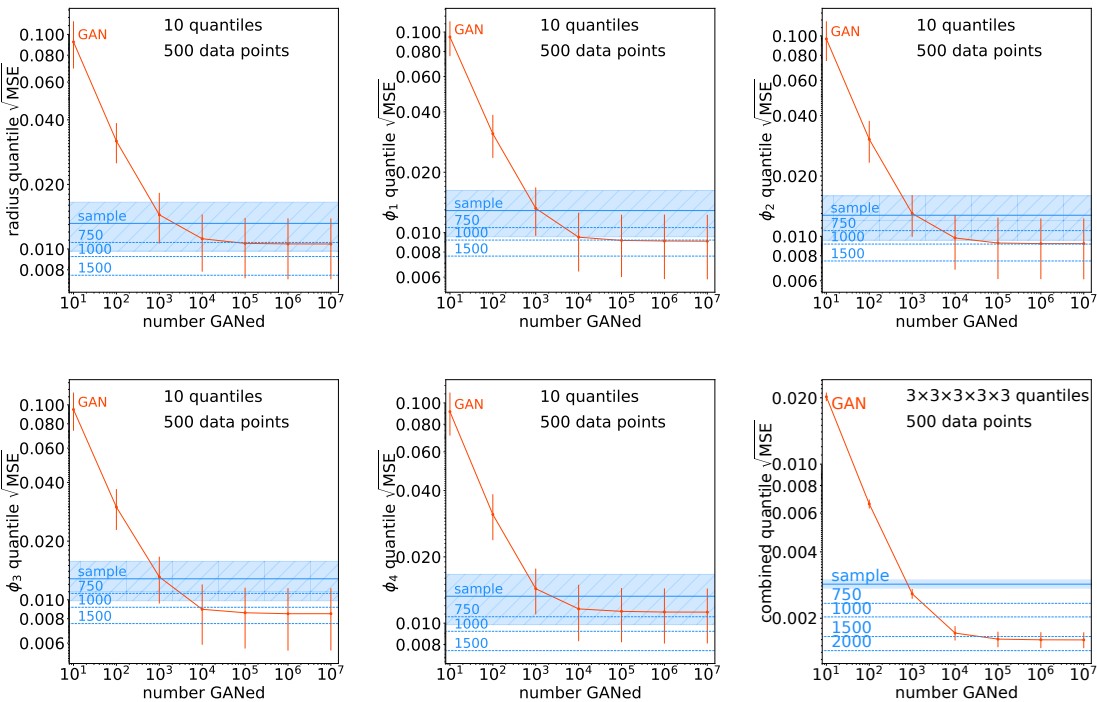

Figure 6: Quantile error for the 5D Gaussian spherical shell for sampling (blue) and GAN (orange). Upper left to lower right we show 10 quantiles projected onto $r$ and the four angular variables, and $3^5$ 5D-quantiles in polar coordinates.

## 4 Multi-dimensional spherical shell

To see the effect of a higher-dimensional phase space we further increase the number of dimensions to five and change the Gaussian ring into a spherical shell with uniform angular density and a Gaussian radial profile

$$P(r) = N_{4,1}(r) + N_{-4,1}(r), \tag{4}$$

with radius $r \geq 0$ and angles $\varphi_{1,..,4}$.

Even if we limit ourselves to the hard scattering, around ten phase space dimensions is typical for LHC processes we would like to GAN [19]. In typical LHC applications, the number of points in a precision Monte-Carlo simulation does not scale with the dimensionality. In other words, in the high-dimensional phase space the nearest neighbors are usually far apart, and without importance sampling it is hopeless to maintain even a modest precision. For our toy example we increase the size of the training sample from 100 to 500 events. One reason is that in a 5D-space 100 data point become extremely sparse; a second reason is that a reasonable 5D-resolution requires at least $3^5$ quantiles, which should be matched by a similar number of training data points.

In Fig. 6 we again show the projected quantiles in the five polar coordinates, starting with the radius. The individual amplification factors range around two for the radius and between 1.3 and above two for the angles. Over the full phase space we define $3^5 = 243$ 5D-quantiles. As for the 1D and 2D cases, the training sample provides on average two events per quantile. In the lower right panel of Fig. 6 we see that the 5D amplification factor approaches four, perfectly consistent with our previous results.

An immediate question, and not unrealistic given the parameters of a typical LHC simulation, is what happens with the GANned events if we require even higher resolution in the

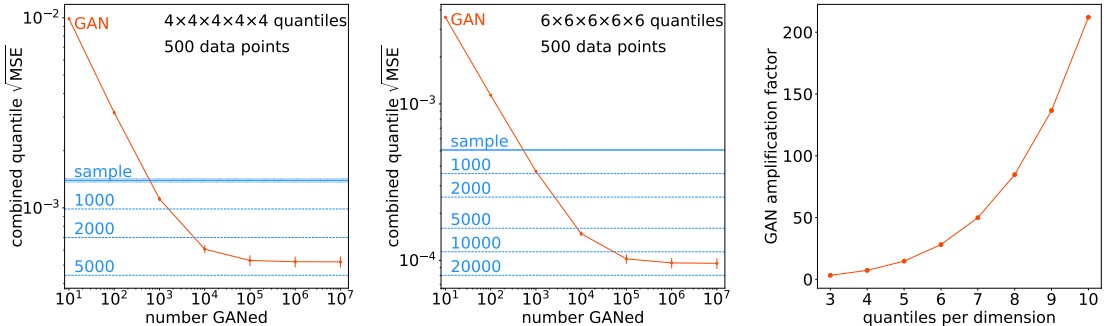

Figure 7: Quantile error for the 5D Gaussian spherical shell $4^5 = 1024$ (left) and $6^5 = 7776$ quantiles, comparing for sampling (blue) and GAN (orange). In the right panel we show the amplification factor as a number of 5D-quantiles.

multi-dimensional phase space. While it is certainly not a situation we appreciate, LHC simulations do contain large phase regions where we expect less than one event in a (training) event sample. In Fig. 7 we see what happens if we increase the number of 5D-quantiles to $4^5$, $6^5$, and beyond. In the left two panels we see that for 6 quantiles per phase space direction we need more than 100.000 GANned events to benefit from an amplification factor around 25. In the right panel we show how this amplification factor scales with a significantly reduced resolution per phase space dimension.

## 5 Outlook

A crucial question for applications of generative models to particle physics simulations is how many events we can generate through the fast network before we exhaust the physics information encoded in the training sample. Obviously, a neural network adds information or physics knowledge though the class of functions it represents. This is the basis for instance of the successful NNPDF parton densities. To answer this question for GANs, we split the phase space of a simple test function into quantiles and use the combined quantile MSE to compare sample, GAN, and, for 1D, a fit benchmark.

We find that it makes sense to GAN significantly more events than we have in the training sample, but those individual events carry less information than a training sample event. As GAN sampling can be much more computationally efficient than *ab initio* simulations, this results in a net statistical benefit. We define an *amplification factor* through the number of hypothetical training events with the same information content as a large number of GANned events. For the 1D-case this amplification factor strongly depends on the number of quantiles or the phase space resolution. While we can never beat a low-parameter fit, the GAN comes surprisingly close and its amplification factor scales with the amplification factor of the fit. For a multi-dimensional phase space the amplification factor for a constant number of training events per quantile is roughly similar. Especially for the kind of multi-dimensional phase spaces we deal with in LHC physics, with their very poor nearest neighbor resolution, neural network interpolation has great potential.

While our focus is on GANs, our observed amplification is also relevant for other applications of neural networks. Specifically, it also applies to networks increasing the data quality through refining [39, 56] or reweighting [57, 58]. It will be interesting to see how the improvement scales with the amount of training data for these approaches.

# Acknowledgments

We thank Mustafa Mustafa, David Shih, and Jesse Thaler for useful feedback on the manuscript. We further thank Ramon Winterhalder for helpful input during the early phases of the project. The research of AB and TP is supported by the Deutsche Forschungsgemeinschaft (DFG, German Research Foundation) under grant 396021762 – TRR 257 *Particle Physics Phenomenology after the Higgs Discovery*. GK and SD acknowledge support by the DFG under Germany's Excellence Strategy - EXC 2121 *Quantum Universe - 390833306*. BN is supported by the U.S. Department of Energy, Office of Science under contract DE-AC02-05CH11231.

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
