# Peer review of "GANplifying Event Samples"

_SciPost Physics, doi:SciPost Phys. 10, 139 (2021)_

## Round 2 · Referee Report · Anonymous (Referee 1) · 2020-12-15

Strengths

This interesting study comes at the right time, answering to the point raised by Matchev and Shyamsundar (Ref. [39] in the manuscript) on the uncertainties associated with using Generative Adversarial Networks as a dataset amplifier tool.

This paper demonstrates that a GAN-based strategy for simulation allows one to obtain a dataset larger than the training dataset size, making the whole process computationally advantageous.

Weaknesses

No major weak point was identified

Report

I believe that the manuscript is easy to read and basically ready to be accepted for publication. I just ask the authors to consider a few remarks (see detailed list) and submit a minor revision.

Requested changes

1- I think that the factor x25 of expected statistics increase should actually be 19. My math is: LHC data amount to 160 fb-1. HL-LHC should deliver 3000 fb-1. So 3000/160 ~ 19. To get you 25, One would need LHC data to amount to 120 fb-1. 2- I would add a reference to "challenging the current simulation tools". For instance the CMS or ATLAS upgrade TDRs. 3- In a few points (e.g., end of section 2), the paper refers to the fact that using GAN is a trade off between statistical and systematic precision of the interpolation applied by the GAN. This trade off is the essential point of this paper, which determines the saturation observed by the carried-on experiments. I think that this aspect should be expanded in the introduction. This fact should be spelled out more clearly, in my opinion, for the benefit of non-expert readers. 4- The next-to-last paragraph in page 2 is a key point to explain why GANs work. I would integrate this paragraph with some reference, if any. You do refer to related aspects in the next paragraph (e.g., video games). On the other hand, these are qualitative statements. Not sure if something more quantitative exists. If not, it should be stated. 5- I am concerned by the fact that 100 points are not a lot, and the fit in Fig. 1 shows some bias (within statistical uncertainties, of course). Did you verify that the fit is unbiased in average? Did you try more options that your 100-point experiments? I am worried that the results of Fig.1 might be biased by the low-statistics of the fit sample. 6- The discussion in section 2 should be integrated with more details, defining clearly what sample, fit and GAN are. The information is there, but scattered across the paper and this makes the reading and understanding more difficult. 7- Could you a legend to Fig. 1 and explain (e.g., in the caption) the meaning of the horizontal lines at 200 and 300, detailing how you computed them? Could you add the 1000 line as well? 8- You say at page 4 that your uncertainty evaluation for the fit is equivalent to using the full covariance matrix returned by the fit. Did you verify that? Why not just using the covariance matrix? I would be interested to see the comparison. I agree with you that this is true in absence of fit biases, for perfectly chi-sq distributions. But are you in that situation? You likelihood should have a two-fold ambiguity and the two minima might overlap if the parameter determination is not precise enough [here I am assuming that the bit parameters are the two mu and sigma + the relative fraction] 9- How was the GAN architecture chosen? Was any optimization performed? Did you try alternative architectures? 10- In the 2D example, why did you centre one of the two Gaussians at a negative r value? Doing so, you reduce this contribution to a tail below the other Gaussian, so the sense of the camel back shape is gone. I was puzzled by this choice. I would have used some other positive value of mu for the second Gaussian. The same comment applies to the N-dim case. 11- In the outlook section, I think that "a neural network ... they represent" should be "it represents".

  • validity: high
  • significance: high
  • originality: good
  • clarity: high
  • formatting: good
  • grammar: good

Author:  Sascha Diefenbacher  on 2021-03-26  [id 1336]

(in reply to Report 1 on 2020-12-15)

First of all, thank you for the remarks, which should indeed make our paper more readable.

  1. We had based this on the CERN website    (https://home.cern/resources/faqs/high-luminosity-lhc) which states that the HL-LHC    target is 4000 fb^-1

  2. Thank you for the suggestion, we have added the requested references in the updated version.

  3. We have added a brief discussion to the introduction.

  4. We are not aware of such an analysis in the ML literature and now    state this explicitly.

  5. We have generated a very sizable number of versions of Fig.1,    without observing a common bias. For larger training samples we see    a similar effect, but of course less dramatic. We now mention this    in the text.

  6. We include this information more visibly now.

  7. We updated the figures to be more readable now, and updated the caption to explain the dotted lines.

  8. Thanks for pointing this out, we have amended the sentence to be more clear, as it is indeed not just the fit uncertainty, but also the variation stemming from the individual dataset. The main reason we chose this method is to make sure we are consistent between our 3 approaches.

  9. We performed some small by hand optimization, mostly varying the learning rates and the depth of the networks. We did also test other setups such as W-GANs and GANs without gradient penalty. However once we found a method that produced consistent and stable training results we did not further optimize the setup, as to keep our message as general as possible.

  10. The reason we use this specific formula is somewhat involved. In principle what we wanted is a Gaussian located at +4 that we can use as our radius. We sample form this Gaussian, then multiply the radii with vectors sampled from a N-dimensional unit sphere and the end result is our N-dimensional data. However a simple Gaussian at +4 (N(4,1)) with a cutoff at 0 is not normalized. This lead us to use the absolute |N(4,1)| This is, however equivalent to the Expression N(4,1) + N(-4,1), as our dateset is by design symmetrical around the origin. Meaning it is irrelevant whether a half the unit sphere vector are assigned a positive radius and the other half a negative one, or if all or assigned a positive radius, the resulting distributions are equivalent. The reason we went with this expression over |N(4,1)| was because we hoped it would make it easier to see the similarities between 1-D and N-D.

  11. Good catch, we changed this.

---

## Round 2 · Referee Report · Veronica Sanz (Referee 2) · 2021-2-2

Strengths

This work explores the idea that event generation by generative networks encodes more statistical knowledge than one would naively expect.

This is an extremely interesting idea, and the paper goes some way to quantify the power of amplification when using GANs in event generation.

Weaknesses

The only weakness I find in this paper is the level it is aimed to. Despite working on related subjects, as a referee I had a bit of a hard time understanding all the statements in the paper, and were left hungry for more explanations to some statements made.

I do suggest below some possible improvements in the write up to improve readability (physics-wise, not English).

Report

I do strongly recommend its publication.

Requested changes

  1. In the Introduction, the authors state that NNs go beyond naive interpolation due to the structure of the network, which e.g. introduces some level of smoothness in the functions it tries to represent. That statement seems intuitive, but one is left with the question of to what extent it holds for different NN architectures and datasets. For example, could the authors comment on why choosing this particular camel back example, and whether they have tested their procedure with other functional forms?

  2. In Sec 2, discussion on Fig 2, the authors state 'the agreement between the sample or the fit and the truth generally improves with more quantiles', namely as we go left to right in Fig. 2. Could the authors clarify this point? It isn't straightforward when inspecting the figure.

  3. In Sec 2, could the authors explain in a bit more detail why mode collapse is expected for this architecture and dataset, and how the tweak of per-batch statistics helps. Is this a generic problem and solution for this kind of analysis?

  4. When moving from 1D/2D (Figs 2 and 4) to higher dimensional examples (Figs. 6 and 7), the number of datapoints goes from 100 to 500. How much do these choices impact the conclusions reached by the authors? In the conclusions we read a bit of discussion on this, but I would like to see something more quantitative.

  • validity: high
  • significance: top
  • originality: high
  • clarity: ok
  • formatting: excellent
  • grammar: excellent

Author:  Sascha Diefenbacher  on 2021-03-26  [id 1335]

(in reply to Report 2 by Veronica Sanz on 2021-02-02)
Category:
answer to question
correction

Thank you, Veronica, for the kind review and for the very good points.

  1. We have played with other functional forms. The main issue was to avoid Gaussian-like functions, because they can be trivially learned. The camel back seemed as un-Gaussian as possible, and its topology with a central hole adds a challenge to the network.

  2. Thank you for pointing this out, we have clarified it in the updated version.

  3. We added a brief explanation in the updated version.

  4. Again, we added a brief comment.

---

## Round 3 · Referee Report · Veronica Sanz (Referee 2) · 2021-4-27

Report

I am happy with the changes made in the text after the referee 1 and myself made recommendations. I recommend publication in its current form.

---

## Round 3 · Referee Report · Anonymous (Referee 1) · 2021-5-19

Report

I am satisfied with the answers provided by the authors and I recommend that the work is accepted for publication

---

## Round 3 · Author Response

We are thankful to the referees for their insightful and helpful comments and suggestions. We integrated the suggestions into the text, with any modifications indicated in the list of Changes. Additionally we would like to clarify some of the raised points:

  • I think that the factor x25 of expected statistics increase should actually be 19. My math is: LHC data amount to 160 fb-1. HL-LHC should deliver 3000 fb-1. So 3000/160 ~ 19. To get you 25, One would need LHC data to amount to 120 fb-1.

->  We had based this on the CERN website   (https://home.cern/resources/faqs/high-luminosity-lhc)  which states that the HL-LHC    target is 4000 fb^-1

  • I am concerned by the fact that 100 points are not a lot, and the fit in Fig. 1 shows some bias (within statistical uncertainties, of course). Did you verify that the fit is unbiased in average? Did you try more options that your 100-point experiments? I am worried that the results of Fig.1 might be biased by the low-statistics of the fit sample.

-> We have generated a very size able number of versions of Fig.1, without observing a common bias. For larger training samples we see a similar effect, but of course less dramatic. We further also  now mention this in the text.

  • You say at page 4 that your uncertainty evaluation for the fit is equivalent to using the full covariance matrix returned by the fit. Did you verify that? Why not just using the covariance matrix? I would be interested to see the comparison. I agree with you that this is true in absence of fit biases, for perfectly chi-sq distributions. But are you in that situation? You likelihood should have a two-fold ambiguity and the two minima might overlap if the parameter determination is not precise enough [here I am assuming that the bit parameters are the two mu and sigma + the relative fraction]

-> Thanks for pointing this out, we have amended the sentence to be more clear, as it is indeed not just the fit uncertainty, but also the variation stemming from the individual dataset. The main reason we chose this method is to make sure we are consistent between our 3 approaches.

  • How was the GAN architecture chosen? Was any optimization performed? Did you try alternative architectures?

-> We performed some small by hand optimization, mostly varying the learning rates and the depth of the networks. We did also test other setups such as W-GANs and GANs without gradient penalty. However once we found a method that produced consistent and stable training results we did not further optimize the setup, as to keep our message as general as possible.

  • In the 2D example, why did you centre one of the two Gaussians at a negative r value? Doing so, you reduce this contribution to a tail below the other Gaussian, so the sense of the camel back shape is gone. I was puzzled by this choice. I would have used some other positive value of mu for the second Gaussian. The same comment applies to the N-dim case.

-> The reason we use this specific formula is somewhat involved. In principle what we wanted is a Gaussian located at +4 that we can use as our radius. We sample form this Gaussian, then multiply the radii with vectors sampled from a N-dimensional unit sphere and the end result is our N-dimensional data. However a simple Gaussian at +4 (N(4,1)) with a cutoff at 0 is not normalized. This lead us to use the absolute |N(4,1)| This is, however equivalent to the Expression N(4,1) + N(-4,1), as our dateset is by design symmetrical around the origin. Meaning it is irrelevant whether a half the unit sphere vector are assigned a positive radius and the other half a negative one, or if all or assigned a positive radius, the resulting distributions are equivalent. The reason we went with this expression over |N(4,1)| was because we hoped it would make it easier to see the similarities between 1-D and N-D.

  • In the Introduction, the authors state that NNs go beyond naive interpolation due to the structure of the network, which e.g. introduces some level of smoothness in the functions it tries to represent. That statement seems intuitive, but one is left with the question of to what extent it holds for different NN architectures and datasets. For example, could the authors comment on why choosing this particular camel back example, and whether they have tested their procedure with other functional forms?

-> We have played with other functional forms. The main issue was to avoid Gaussian-like functions, because they can be trivially learned. The camel back seemed as un-Gaussian as possible, and its topology with a central hole adds a challenge to the network.

---

## Round 3 · List of Changes

• Added  reference to 'challenging the current simulation tools' statement

  • Added a brief discussion to the introduction about statistical and systematic precision trade off of GANs.

  • Added mention in text that amplification is also observable for larger sample sizes

  • Made descriptions of the 3 compared approaches (Sample, fit, GAN) more visible

  • Updated the figures to be more readable, and updated the caption to explain the dotted lines.

  • Amended the sentence on fit uncertainties to be more clear

  • Clarified sentence regarding the effects of increasing the numbers of quantiles

  • Added a brief explanation on how we prevent mode collapse in the GAN training.

  • Added a brief comment why we increase the number of samples from 100 to 500 when going to 5D

---

## Editorial Decision

published